# OpenReview forum: "Persona-Assigned Large Language Models Exhibit Human- Like Motivated Reasoning"
_colmweb.org/COLM/2025/Workshop/Social_Sim — Social Sim'25_

### Official Review · Reviewer_p9Ef · 2025-07-13
**Analyzing motivated reasoning in persona-assigned LLMs**

**Rating:** 7
**Overall Assessment:** 4
**Confidence:** 4

**Review:**

The paper is generally well-written with a comprehensive appendix. It positions itself to be the first in using identity-congruent reasoning as a theoretical framework for analyzing persona-induced biases in LLMs. The authors conduct a thorough evaluation across eight personas spanning political, religious, education level and gender categories, and incorporate established cognitive assessments, lending depth to their analysis. Importantly, the paper's findings are aligned with studies on motivated reasoning on human subjects, and the failure of the prompt-based debiasing efforts show that these inherent biases are imbued at much deeper levels in the LLMs. However, the paper seems to tackle a rather limited set of experiments - deliberately excluding certain demographic personas from the second task and restricting their evaluation to small datasets for both tasks. This limited scale raises concerns about the generalizability and robustness of the findings across broader domains or more complex scenarios.

**Comments Suggestions And Typos:**

N/A

**Paper Summary:**

This paper analyzes motivated reasoning as a framework for understanding identity-congruent reasoning in persona-assigned LLMs. The authors assign 8 distinct personas to LLMs on two tasks – veracity discernment of fake vs real news headlines and numerical scientific evidence evaluation. When evaluated against a baseline, OpenAI and open-source LLMs exhibit different effects – persona assignment reduces veracity discernment in models by up to 9%, and political personas are upto 90% more likely to evaluate scientific evidence when the ground truth is aligned with their political ideologies. Further the authors report that prompt-based debiasing techniques such as Chain of Thought and accuracy prompting fail to mitigate biased motivated reasoning.

**Relevance:**

5

**Summary Of Strengths:**

* The paper introduces identity-congruent reasoning as an original framework for analyzing persona-induced bias in LLMs.
* It conducts a rigorous evaluation across both proprietary and open-source models, using eight personas spanning four demographic dimensions.
* It also includes analytical thinking assessments via actively open-minded thinking (AOT) questions and the six-point Cognitive Reflection Test (CRT).
* The demonstrated failure of popular prompt-based debiasing methods has strong implications for the field, highlighting the need to go beyond surface-level prompt mitigation techniques.

**Summary Of Weaknesses:**

* The study only uses political identity for Task 2, omitting potentially impactful variables such as religious identity and education level. These factors could plausibly influence reasoning around scientific claims and deserve investigation.
* The experimental scale is limited: the veracity task uses ~20 headlines, and the scientific reasoning task includes only two experiments. It is unclear whether the observed effects generalize across broader datasets.

---

### Official Review · Reviewer_jbBM · 2025-07-17
**Persona-Assigned Large Language Models Exhibit Human-Like Motivated Reasoning**

**Rating:** 9
**Overall Assessment:** 4
**Confidence:** 4

**Review:**

There is good structure and there is clear separation between the problem motivation, experimental design and analysis. The topic is also a timely contribution since there is a lot of prior work exploring LLM biases and persona effects, but I think the application of motivated reasoning in a theoretical lens has been less explored. There is good use of statistical models to isolate the effects of personas on reasoning performance and the experiments are also large at scale (with over 21000 prompts), including ablations and validation checks. Overall I think that these findings could be useful to consider future work regarding the cognitive evaluation of AI also from an alignment/safety perspective.

**Comments Suggestions And Typos:**

N/A

**Paper Summary:**

This paper investigates the idea of whether there is motivated reasoning in persona-assigned large language models. Motivated reasoning is defined as a cognitive bias where humans reason toward identity-congruent conclusions. The authors explore whether similar behaviors can emerge in LLMs that are prompted to adopt human-like personas across different dimensions such as political, educational, religion, and gender.

The methodology involved using two psychological tasks adapted from human studies to analyze eight LLMs. The main findings were that the persona assignment decreases veracity discernment accuracy by up to 9%. Political personas show up to 90% increased accuracy when evaluating evidence congruent with their identity. Motivated reasoning (as measured via AOT scores) predicts LLM behavior better than analytical reasoning (CRT). And prompt-based debiasing methods such as chain-of-thought were largely ineffective.

**Relevance:**

5

**Summary Of Strengths:**

Interdisciplinary study, perofrmed large scale experiments (across 8 LLMs and 8 personas). There are clear psychological constructs utilized (AOT, CRT, VDA). Raise some good concerns about personalized AI and epistemic bubbles.

**Summary Of Weaknesses:**

It would be good to have some more insight into why say persona prompts induce motivated reasoning. This could be from more mechanistic insight and future work should probe training dynamics or fine tuning procedures. The debiasing section could be developed more as well with more explorations into fine-tuning or the multi turn dialogue strategies. There is no error analysis on why certain personas or models perform worse (such as some of the GPT variants)

---

### Meta-Review · Program_Chairs · 2025-07-24

**Recommendation:** Accept

**Metareview:**

--